# Late Cretaceous ammonoids show that drivers of diversification are regionally heterogeneous

Joseph T. Flannery-Sutherland [1,2] ✉, Cameron D. Crossan[2], Corinne E. Myers [3], Austin J. W. Hendy[4], Neil H. Landman [5] & James D. Witts[2,6]

Palaeontologists have long sought to explain the diversification of individual clades to whole biotas at global scales. Advances in our understanding of the spatial distribution of the fossil record through geological time, however, has demonstrated that global trends in biodiversity were a mosaic of regionally heterogeneous diversification processes. Drivers of diversification must presumably have also displayed regional variation to produce the spatial disparities observed in past taxonomic richness. Here, we analyse the fossil record of ammonoids, pelagic shelled cephalopods, through the Late Cretaceous, characterised by some palaeontologists as an interval of biotic decline prior to their total extinction at the Cretaceous-Paleogene boundary. We regionally subdivide this record to eliminate the impacts of spatial sampling biases and infer regional origination and extinction rates corrected for temporal sampling biases using Bayesian methods. We then model these rates using biotic and abiotic drivers commonly inferred to influence diversification. Ammonoid diversification dynamics and responses to this common set of diversity drivers were regionally heterogeneous, do not support ecological decline, and demonstrate that their global diversification signal is influenced by spatial disparities in sampling effort. These results call into question the feasibility of seeking drivers of diversity at global scales in the fossil record.

Inference of patterns and processes of diversification in the fossil record has a long history of palaeobiological interest[1–4]. Ever larger compilations of fossil occurrence data and improved chronostratigraphic constraints have enabled precise quantification of the magnitude and timing of interspersed mass extinctions and evolutionary radiations[5–7]. In turn, this has permitted investigation of the drivers of diversification, broadly divided into intrinsic regulation of taxonomic richness by diversity-dependent mechanisms (the Red Queen Hypothesis)[8–12] versus extrinsic, abiotic forcings imposed by the Earth-Solar system (the Court Jester Hypothesis)[4,13,14]. The influence of the Red Queen remains contentious due to the difficulty of establishing whether biodiversity has ever truly entered a curtailed, diversity-dependent regime[15–20], with the additional observation that interactions between life and the Earth-Solar system through geological time have dynamically altered Earth's carrying capacity[21,22]. By contrast, a series of Court Jester mechanisms are repeatedly hypothesised to have driven patterns of diversification in individual clades to entire biotas, for example, the linked effects of atmospheric $CO_2$ concentration and temperature, the configuration of the continents, or eustatic sea-level variations[19,23–29].

[1]School of Geography, Earth and Environmental Science, University of Birmingham, Birmingham, UK. [2]Palaeobiology Research Group, School of Earth Sciences, University of Bristol, Bristol, UK. [3]Department of Earth and Planetary Sciences, University of New Mexico, Albuquerque, NM, USA. [4]Natural History Museum of Los Angeles County, Los Angeles, CA, USA. [5]Division of Paleontology (Invertebrates), American Museum of Natural History, New York, NY, USA. [6]Department of Earth Sciences, Natural History Museum, London, UK. ✉e-mail: j.t.flannerysutherland@bham.ac.uk

It is also widely recognised that diversity patterns in the fossil record are skewed by geological and anthropogenic biases[1,6,30–33], fuelling development of increasingly sophisticated methods for quantifying diversification dynamics from incomplete, biased fossil occurrence data. In the last decade, Bayesian approaches, which couple birth-death and preservation processes have enabled estimation of sampling-corrected origination and extinction rates from fossil occurrences[34–37], avoiding the problems of inferring these fundamental rates from extant phylogenies[38–42]. In turn, lineage birth and death rates can be modelled as functions of their potential drivers[43–45], permitting separate consideration of the factors that promoted origination or drove extinction. These models have enjoyed widespread uptake by the palaeontological community[46–61] but the paradigm of global biodiversity analysis, whether of individual clades or entire biotas, is challenged by the observation that the spatial distribution and extent of the fossil record varies through geological time[31,32,62]. This spatial variation induces geographic sampling biases that distort our view of taxonomic richness and diversification rates even after correction for geological sampling biases[31]. Not only is it inadvisable to treat the fossil record as a representative sample of varying global diversity, with some workers questioning whether global patterns are biologically informative in the first place[63], but this also overlooks its well-established biogeographic nuances, necessitating spatially sensitive approaches[31,32,64–67].

Given that diversity shows spatial heterogeneity, it is reasonable to also expect this of its drivers and such variation is evident in the present-day biosphere, for example latitudinally structured co-variation in irradiance, climate and species richness[68–70]. Some palaeontologists, however, have continued to compare 'global' fossil records to potential drivers without considering regional heterogeneity in diversification processes[64,71,72]. Here we investigate diversification dynamics of Late Cretaceous ammonoids in a regionalised framework (Fig. 1) to determine whether the drivers of diversity show this expected variation. Ammonoids were an iconic clade of pelagic shelled cephalopods, which originated in the Devonian and survived multiple mass extinction events up until their demise at the Cretaceous-Paleogene (K-Pg) boundary[73,74]. Late Cretaceous ammonoids provide an ideal study system for our investigation as their fossil record is taxonomically mature and extremely well-sampled compared to many other fossil clades[75]. Ammonoids additionally present an intriguing controversy regarding whether they were in ecological decline prior to their definitive extinction at the end of the Cretaceous[73,76,77], with previous authors highlighting the importance of combining global and regional analyses to tackle this problem[78–80]. Palaeontologists are also rapidly recognising the wealth of 'dark data', present in museum collections but unrecorded in published literature, that has the potential to transform the scale and completeness of palaeobiological analyses[66,81–86]. We, therefore, take the opportunity to unite publicly available fossil occurrence data with 'dark' datasets to address this emerging gap in both knowledge and practice.

Here we show that a regionally heterogeneous interplay of Red Queen versus Court Jester processes underpinned spatial variation in ammonoid diversification in the Late Cretaceous. These results highlight the challenges that face confident inference of the drivers of diversity in the geographically biased fossil record and call for more nuanced consideration of spatiotemporal complexity of the processes that jointly underlie past biodiversity and its present-day geological remnants.

## Results
### Global ammonoid diversification dynamics
We captured and spatially standardised regional samples of fossil occurrence data for the Antarctic, Tethys, Western Interior Seaway (WIS), Atlantic and Gulf, East Pacific, West Pacific, West Africa, and South Africa (Fig. 1), then estimated genus and species-level origination and extinction dynamics from these data in addition to global dynamics (Fig. 2), using a birth-death-sampling model implemented in a Bayesian framework (PyRate). In all analyses, species-level origination ($\lambda_s$) and extinction ($\mu_s$) rates were more volatile compared to the genus-level rates ($\lambda_g$, $\mu_g$), leading to sharper fluctuations in taxonomic diversity (Supplementary Figs. 1–9). The differential quality of the genus and species identifications for the same set of fossil occurrence data led to drops in species richness below genus richness during some intervals in the regional analyses (Fig. 3), even though the former should minimally equal the latter (i.e., at least one species present per genus), highlighting a limitation of our data, but ultimately patterns of diversity at both taxonomic levels were qualitatively similar in all analyses (Fig. 3).

At the global level, $\mu_g$ was broadly stable through the Cenomanian to Santonian stages, punctuated by a spike at end of the Cenomanian (Fig. 2A). A further spike occurred in the early Campanian, followed by a reduced rate for the remainder of the stage. $\mu_g$ then intensified gradually through the Maastrichtian before spiking at the end of the Cretaceous (Fig. 2A). The end-Cenomanian spike, Maastrichtian intensification and end-Cretaceous spike were mirrored by $\mu_s$ but were accompanied by additional pulses at the ends of the Turonian, Coniacian, and Santonian, with stepwise decreases in the intervening background rate (Fig. 2A). $\mu_s$ then increased during the early Campanian to an elevated background rate, although the uncertainties surrounding this rate increase indicate the potential for a much more prominent spike during this interval. $\lambda_g$ remained stable through the Cenomanian to the middle Coniacian until a sudden drop, then showed continued stability for the remainder of the Cretaceous (Fig. 2A). $\lambda_s$ displayed pulses during the late and terminal Coniacian, preceding reduction to a slightly lower, but stable rate from the early Turonian to the end of the Coniacian (Fig. 2B). $\lambda_s$ then followed a pattern of volatile spikes and crashes through the early Cenomanian, a final rapid intensification in the latest Cenomanian, and gradual decline through the Maastrichtian (Fig. 2B).

The outcome of these rates was a broadly stable pattern of genus richness through the Cenomanian to Santonian punctuated by losses at the beginning of the Turonian (coincident with Ocean Anoxic Event 2–OAE2) and Coniacian, a phase of decline through the early Campanian, then a second interval of stability until the end-Cretaceous mass extinction (Fig. 2D). At the species level from the Cenomanian to Santonian, diversity dropped suddenly at the beginning of the stage followed by a more gradual recovery, leading to a pattern of stepwise decline through this interval (Fig. 2D). Species richness crashed successively at the start and middle of the Campanian, with the intervening time displaying high levels of uncertainty in global diversity estimates, recovered suddenly in the early Maastrichtian, then declined in the late Maastrichtian prior to extinction at the K-Pg boundary (Fig. 2D).

### Regional ammonoid diversity trends
Spatially varying diversification rates and diversity trajectories demonstrate that 'global' ammonoid dynamics in the Late Cretaceous were heterogeneous between regions and taxonomic levels (Figs. 2 and 3). There is a general and expected trend of increasing diversity with region size under the species-area effect, although the WIS is the most diverse region despite being smaller than Tethys (Fig. 2E, F). Qualitatively, the global trend is primarily a synthesis of patterns from the larger, historically well-sampled regions within the Global North (Tethys, WIS, Atlantic and Gulf, East Pacific), although these regional patterns still display their own nuances (Fig. 2E, F). Notably this correspondence between the global curve and specific regional trends is still apparent despite the former including data that was not present in any of the other regions, for example, more sporadically sampled occurrences from South America, Australia and Madagascar (Fig. 1).

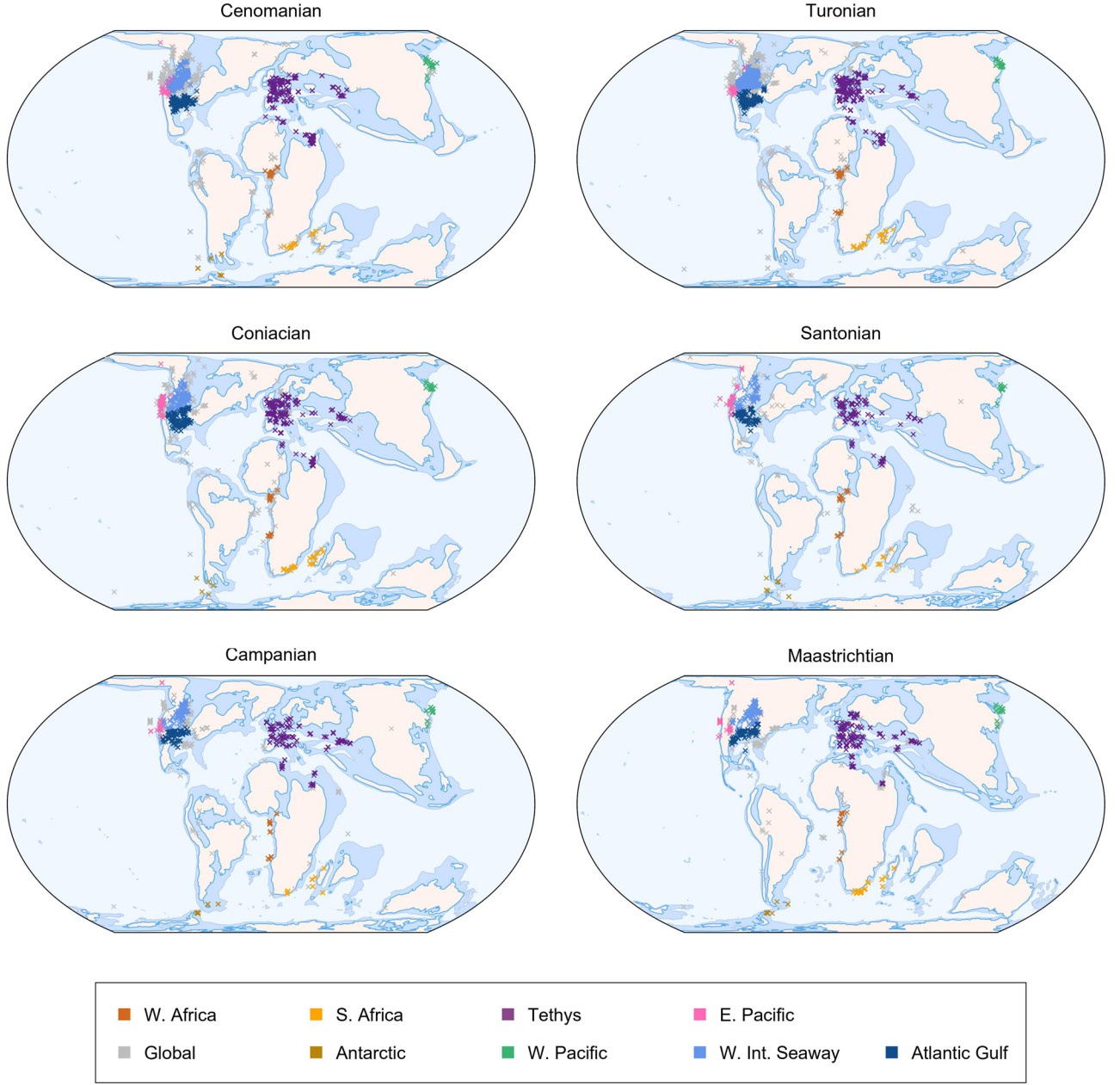

**Fig. 1 | Stage-level ammonoid sampling regions in the Late Cretaceous.** Chosen regions corresponding to biogeographically distinct ammonoid provinces with sufficient data to derive spatially standardised subsamples for reliable estimation of diversification dynamics. The global dataset encompasses the grey points in addition to all coloured regional points. Source data is available in the electronic supplement accompanying this paper.

Declines in genus and species richness occurred during the interval containing OAE2 in Tethys, Atlantic and Gulf, WIS, and East Pacific, but the onset of declining species richness was earlier in the latter (Fig. 3F). Trends outside of the Global North were more varied, with genus richness increasing substantially in West Africa through the Cenomanian (Fig. 3E), but only marginally in the West Pacific and Antarctic (Fig. 3A, G), while South Africa experienced no change (Fig. 3C). Genus richness then declined from the late Cenomanian into the early Turonian in the West Pacific, while West Africa and the Antarctic experienced minor diversity losses during OAE2, and South Africa instead experienced a rapid increase across the same time interval. Sharp species loss took place in all four regions in the Global North at the end of the Turonian, but corresponding loss of genera did not take place in the WIS (Fig. 3D). Declines in species and genus richness also took place in South Africa, West Africa and East Pacific, while the Antarctic displayed little to no change across this boundary.

In South Africa, this led to continued decline into the Campanian while the other regions showed recovery during the intervening Coniacian and Santonian stages. Generic richness declined gradually in the WIS, Tethys and the Atlantic and Gulf at the beginning of the Campanian, while the East Pacific experienced a much sharper crash several million years later (Fig. 3F). Species-level dynamics at this time were even more varied, with a gradual decline in Tethys, a sharp spike then equally sudden crash in the East Pacific and the Atlantic and Gulf, and turbulent, highly uncertain fluctuation in the WIS (Fig. 3B, D, F, H). Recovery of diversity varied in its intensity across the Global North in the late Campanian and Maastrichtian, with Tethys displaying the greatest increase in species richness and most stable genus richness through the latter interval (Fig. 3B). Away from the Global North, genus richness also recovered towards the end of the Cretaceous (Fig. 3A, C, E, G), but this recovery began earlier in the Antarctic and West Pacific in the late Campanian, compared to the Maastrichtian in South and West Africa.

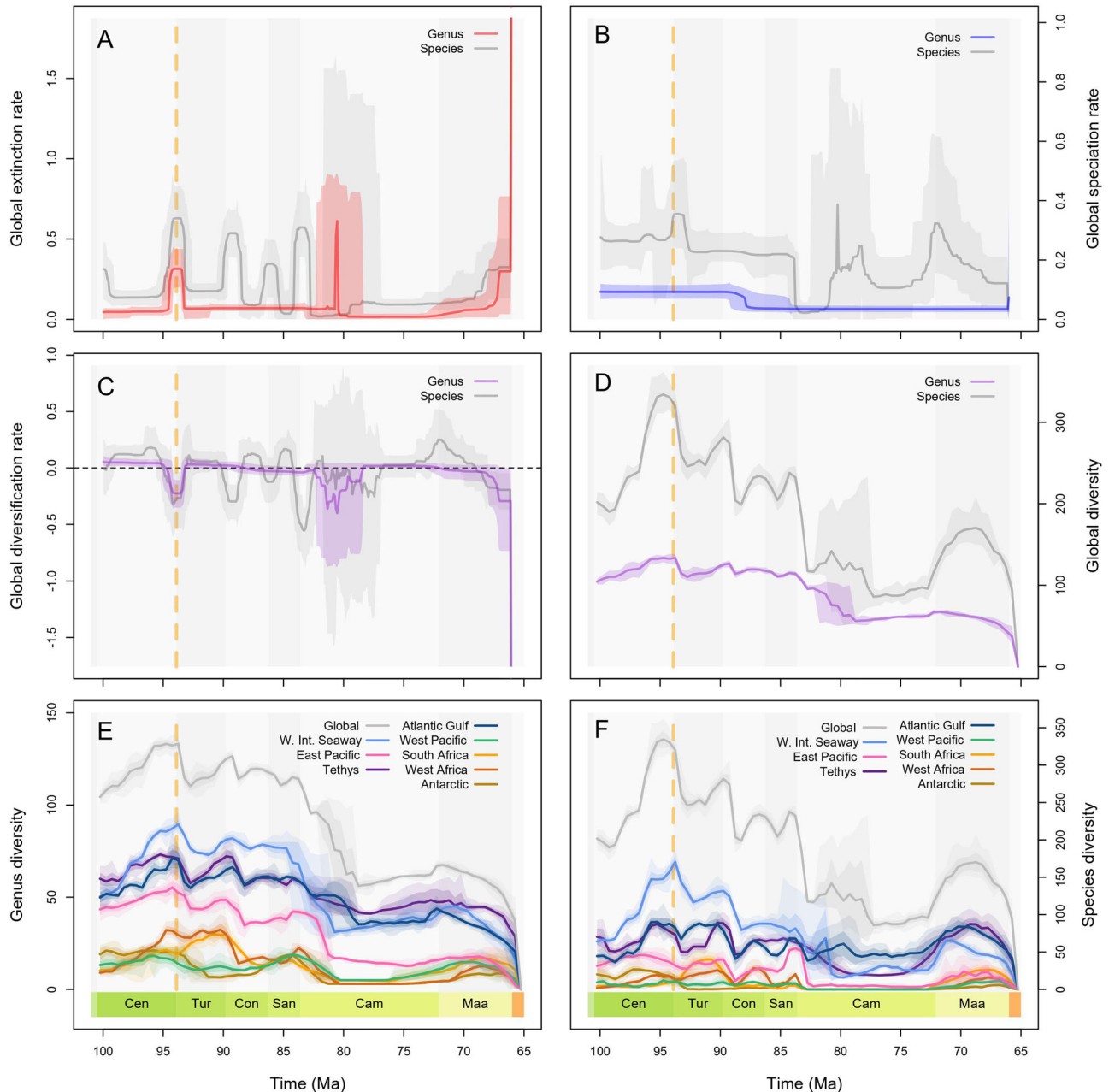

**Fig. 2 | Global ammonoid diversification dynamics in the Late Cretaceous.** Diversification dynamics (mean + highest posterior density estimates−HPDs) were estimated at genus and species levels from spatially standardised data in a Bayesian framework that accounts for temporal sampling biases. Ocean Anoxic Event 2 is marked in each panel by the dashed yellow line. **A** Global mean extinction rate and 95% HPD estimate. **B** Global mean origination rate and 95% HPD estimate. **C** Global mean diversification rate and 95% HPD estimate. **D** Global mean genus and species richness and 95% HPD estimates. **E** Regional mean genus richness with 75% and 95% confidence intervals. **F** Regional mean species richness with 75% and 95% confidence intervals. Source data is available in the electronic supplement accompanying this paper.

Despite their greater volatility, trends in species richness in these regions generally support the trends recovered at the genus-level (Fig. 3).

**Biotic and abiotic drivers of diversification**

We used a multivariate birth-death model (PyRateMBD) to identify time-continuous correlates of our regional origination and extinction rates (Fig. 4). Biotic factors considered were the effects of ammonoid diversity on their own origination and extinction rates (diversity dependence), the influence of food availability (based on nanno-plankton and planktic foram diversity) and the pressure of pelagic predators/competitors (based on ichthyolith abundance), while

abiotic factors were long-term changes in sea level, atmospheric $CO_2$ concentration, and sea surface temperature. At the global level, $\mu_g$ and $\mu_s$ showed significant positive relationships with long-term sea level (+ve), significant negative relationships with ichthyolith abundance and sea surface temperature, $\mu_s$ displayed significant positive relationships with ammonoid and planktic foram diversity, and a significant negative relationship with atmospheric $CO_2$ concentration.

At a regional scale, species-level rates generally displayed a greater number of significant relationships with our chosen drivers compared to genus-level rates (Fig. 4), with significant relationships also becoming more common in the smaller sampling regions. Extinction rates at both taxonomic levels most consistently showed

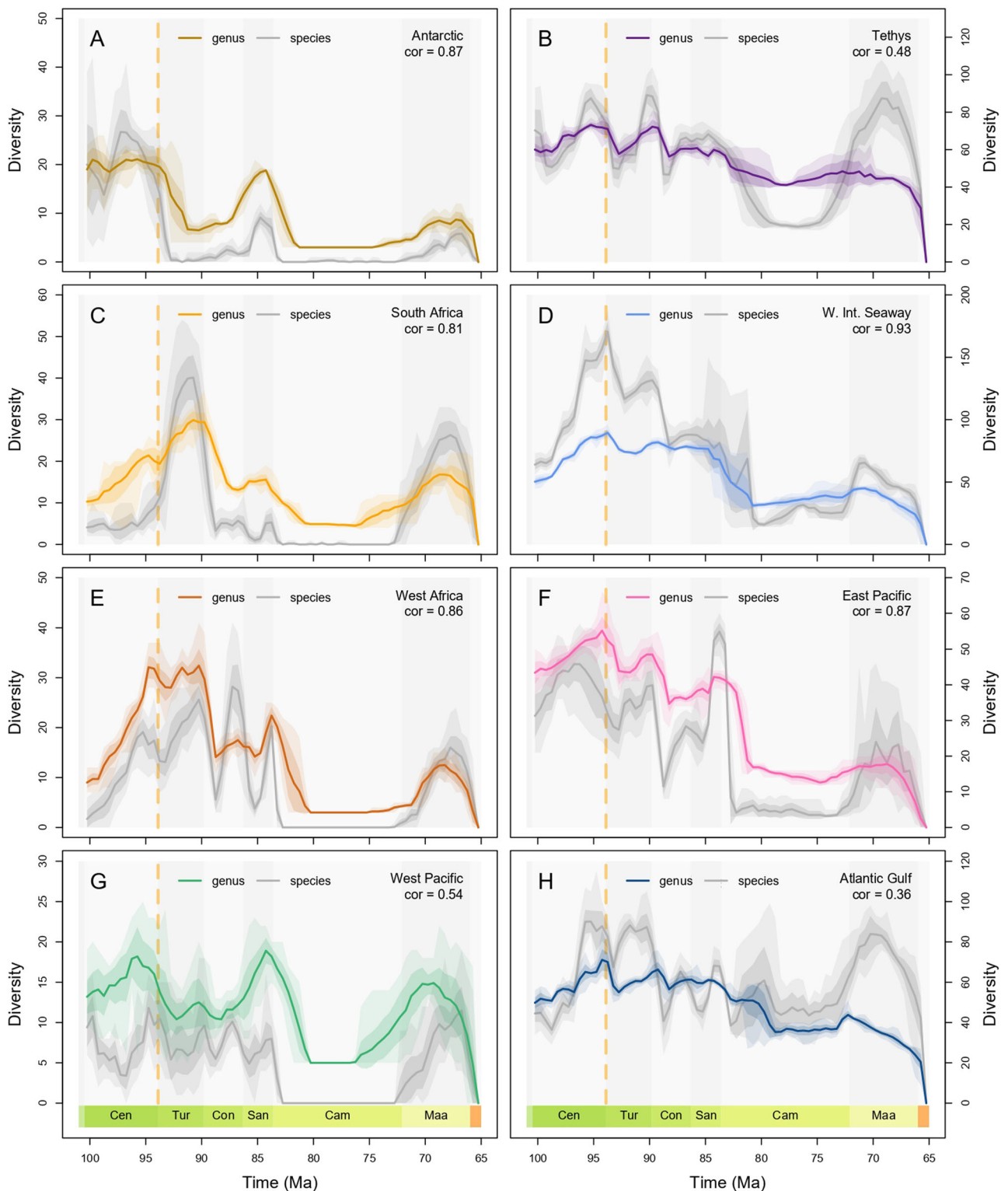

**Fig. 3 | Regional ammonoid diversity in the Late Cretaceous.** Diversity trajectories comprising the mean and 75% and 95% confidence intervals for each regional ammonoid dataset, and Pearson's correlation coefficient between genus and species richness through time. Logically, genus richness should not exceed species richness. This artefact is the result of incomplete sampling of genus richness at the species level in our data, and we direct the reader to the discussion for a more detailed treatment of this issue. **A** Antarctic. **B** Tethys. **C** South Africa. **D** Western Interior Seaway. **E** West Africa. **F** East Pacific. **G** West Pacific. **H** Atlantic and Gulf. Source data is available in the electronic supplement accompanying this paper.

significant negative and positive relationships with sea surface temperature and long-term sea level, respectively, aside from at the genus level in Antarctica, where the relationship was instead negative. Ammonite diversity also displayed recurring significant negative

relationships with origination rates at the species level and the genus level to a lesser extent. Beyond these broad trends, however, drivers of diversification showed complex variation between regions and taxonomic levels, encompassing cases where relationships remained

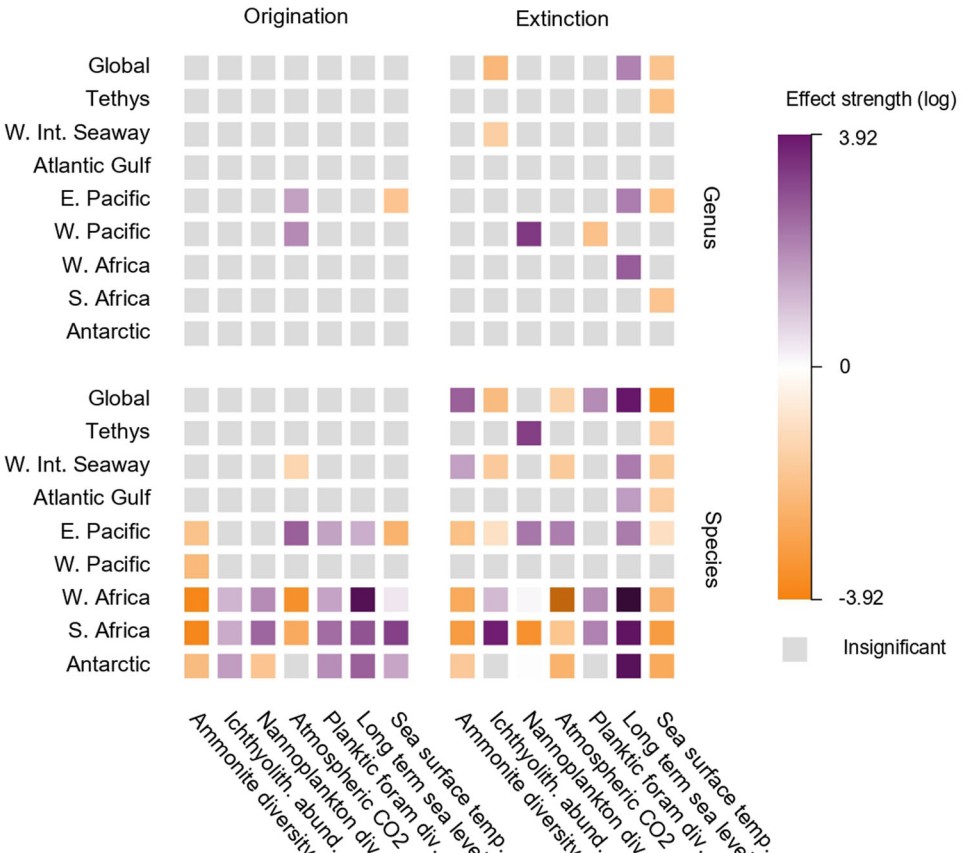

**Fig. 4 | Biotic and abiotic drivers of ammonoid diversification rates in the Late Cretaceous.** The effects of abiotic and biotic drivers on global and regional ammonoid origination and extinction rates inferred from a multivariate birth-death model at genus and species levels. div. = diversity; abund. = abundance; temp. = temperature. Source data is available in the electronic supplement accompanying this paper.

significant but changed their sign or instances where significance changed entirely (Fig. 4).

### Taxonomic predictors of extinction risk

Alongside time-continuous predictors, we also used a multi-trait extinction model (PyRateMTE) to investigate the effects of discrete traits on regional ammonoid extinction risk (Fig. 5). Rather than define and measure a series of ecologically meaningful traits across all Late Cretaceous ammonoid taxa, a task that may not even be possible for some taxa depending on the quality of their fossils, we instead used their suborder and superfamily classifications as these categories are considered to represent distinct ecomorphological guilds which summarise a range of taxonomically diagnostic, functional ecological characteristics[83–85]. At the global scale, suborders displayed predictive power for both $\mu_g$ and $\mu_s$, while superfamilies were only predictive for the latter (Fig. 5). Similarly, suborders displayed greater predictive power for regional $\mu_g$ and $\mu_s$ with significant effects present in four regions, compared to two and three regions for superfamilies at the genus and species levels respectively. Predictive power varied regionally depending on the taxonomic level of the extinction rates under consideration and on the predictor clades themselves, although the Perisphinctina showed the greatest vulnerability across regions and taxonomic levels (Fig. 5).

### Discussion

Previous work has disputed whether ammonoids experienced ecological decline preceding their total extinction at the end of the Cretaceous[73,86]. Globally, species richness fluctuated through the first half of the Late Cretaceous but was ultimately higher at the end of the Santonian than at the beginning of the Cenomanian, while genus richness remained largely stable (Fig. 2E, F). The decline was apparent at both levels at the beginning of the Campanian but was followed by recovery in the Maastrichtian, particularly at the species level (Fig. 2F), which refutes previous suppositions of progressive decline through the Late Cretaceous[77]. The trends displayed in the global curves are distorted by uneven sampling intensity between regions, however, with a bias towards Tethys and the WIS (Fig. 2E, F). Instead, the regional decomposition of global curves demonstrates the spatial heterogeneity of ammonoid diversification dynamics, congruent with findings elsewhere in the fossil record[31,65,87]. Diversity peaked at different times between different regions, with the only broadly consistent diversity patterns across regions comprising declines around the time of OAE2 excepting South Africa, declines at the Turonian-Coniacian boundary which were muted in the WIS, an early Campanian diversity minimum, and the total loss of diversity in all regions at the end of the Cretaceous. Otherwise, ammonoid diversity patterns were regionally distinct (Fig. 3) and did not follow an overarching trend of progressive decline through the Late Cretaceous.

The intensity of ammonoid diversity fluctuations varied regionally through globally felt long-term environmental shifts, including the Cretaceous Thermal Maximum and peak Phanerozoic sea level around 90 million years ago, followed by cooling and eustatic drops through the remainder of the period[88–90]. Heterogeneity was also present during individual geological events, most notably OAE2, a carbon cycle perturbation and pervasive oceanic hyperthermal driven by Large Igneous Province volcanism associated with a global expansion of oxygen-deficient conditions and coincident with elevated extinction rates in a variety of clades[91,92]. Spatial and temporal variation in the development and intensity of low oxygen conditions across OAE2[93], as well as ongoing debate about the extinction mechanisms and

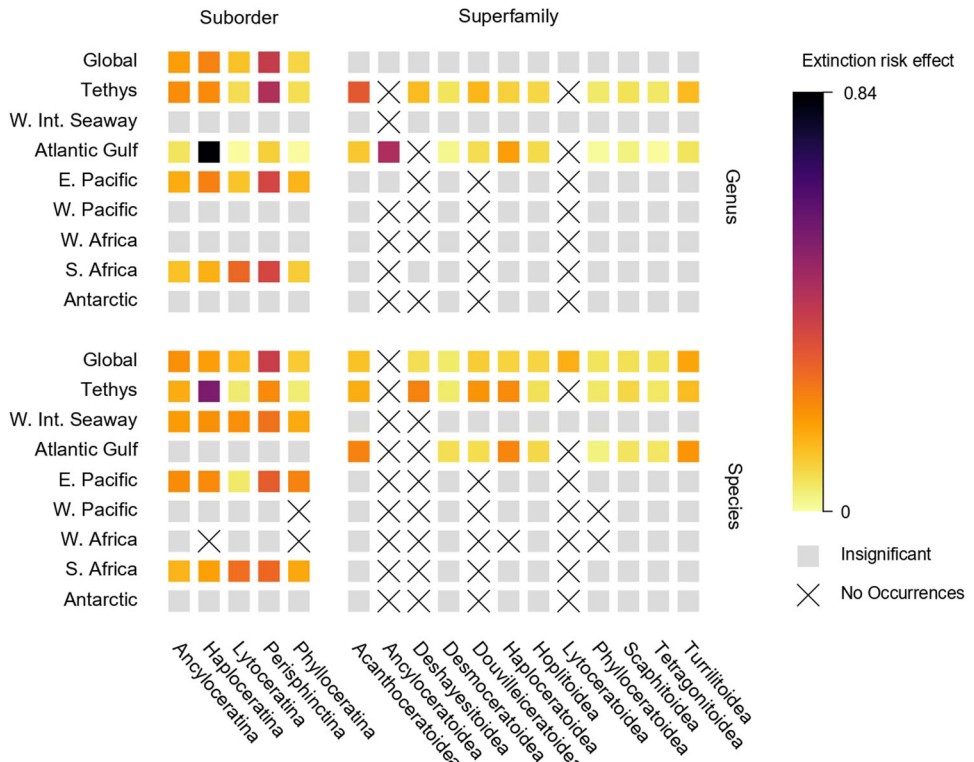

**Fig. 5 | Taxonomic predictors of ammonoid extinction risk in the Late Cretaceous.** The predictive effects of suborder and superfamily classification on ammonoid lineage extinction risk inferred from a multi-trait extinction model. Cells with crosses denote cases where there were no occurrences for a given clade in a region, either due to genuine absence or a lack of occurrences classified at genus or species level. Source data is available in the electronic supplement accompanying this paper.

ecological impact[92] indicate that environmental change associated with this event was likely regionally heterogeneous. In turn, regionally varied response of ammonoids across the Cenomanian–Turonian boundary interval in our data (Fig. 2E, F) match previous suggestions of variation in both the timing and rate of turnover during OAE2[78,80] and support broader findings that the impacts of extinction events were spatially variable[31,65].

Two factors emblematic of Court Jester dynamics, sea level and sea surface temperature, showed relatively consistent negative and positive effects respectively on extinction rates across regions (Fig. 4). Rising sea level correlating with increased ammonoid extinction is unusual at first glance as regression-driven loss of habitat space is more commonly associated with elevated extinction[94,95]. This effect is potentially explicable, however, by the coincidence of elevated extinction rates during OAE2 with continued sea level rise to the Phanerozoic eustatic peak around three million years later, while other studies have also noted that the relationship between sea level change and habitat extent is spatially complex[96]. As such, the effects of regression on speciation and extinction rates may not be straightforward to identify at global scales using a single sea level curve as we have attempted here, although palaeobiological analyses continue to support that the effect of sea level on diversification is not negligible[28,95,96]. Cooling temperatures positively affecting extinction is more congruent with previous inferences regarding ammonoid diversity patterns[78,97]. Notably, modern cephalopods can compensate physiologically for sudden temperature change and may be more sensitive to the combined effects of increasing temperature under hypoxic conditions, but ubiquitous negative effects of ocean warming across cephalopod lineages and ontogenies remain clear[98]. Our analyses do not account for the combined effect presented by temperature-dependent hypoxia, but conversely it is unknown as to how their tolerances may have differed from those observed in modern coeloid cephalopods, although current evidence supports similarity in their basic organismal biology[99].

Conversely, the negative effect of ammonoid diversity on their origination rates is indicative of Red Queen dynamics controlling their macroevolutionary trajectory. These effects appear strongest at smaller spatial scales, however, given the prevalence of significant correlations in the smaller sampling regions and their near absence in the larger (Fig. 4). This fits with previous assertions that Red Queen effects are stronger over smaller spatial scales and are overwhelmed by Court Jester effects at broader spatial scales[4,100]. Beyond these factors, regional ammonoid origination and extinction rates responded variably to our common set of biotic and abiotic drivers. These results call into question previous studies which have identified links between a plethora of drivers and the total diversities of a variety of clades and biotas through the Phanerozoic[46–61]. Given the correspondence between 'global' diversity dynamics and large, historically well-sampled regions, its inferred drivers may conceivably be distorted by geographic sampling biases and so primarily reflect diversity dynamics in only certain parts of the globe. Our global extinction correlations are most closely matched at the species and genus levels by the correlations in the WIS, Tethys, Atlantic and Gulf, and East Pacific (Fig. 4). As with their diversity trends, however, these regions display idiosyncratic correlations that diverge from the global trend, and smaller regions show still greater variability in their responses to Court Jester drivers. Similarly, the positive and negative correlations between global species extinction rates and ammonoid diversity and ichthyolith abundance respectively changed their signs in the smaller regions, highlighting differential responses to Red Queen effects.

Not only is analysis of the relationship between a candidate driver and a spatially incomplete fossil record inadvisable without considering the nuances of its regional composition, but there is also no guarantee that the state of candidate driver accurately captures its

global average. For example, the seminal sea level curves of Haq and its derivations[101–104] are demonstrably biased by regional tectonic architecture and sedimentation affecting sequence stratigraphy, leading to sea level records that more accurately reflect eustatic variation in only certain parts of the globe[105,106], yet these curves are still frequently used in long-term palaeobiological analyses of diversity (e.g., Lehtonen et al.[45]). Similarly, curves of long-term global temperatures from a variety of lithological proxies are biased by temporal variation in their palaeogeographic extent and distribution[107]. As such, it is doubly inappropriate to compare regionally biased diversity records to regionally biased drivers when investigating Court Jester dynamics in geological past. This is also a limitation for our analyses as we compare our robust regional diversification records to potentially quasi-global drivers derived from spatially biased data, although we attempted to choose driver curves which were based on reliable proxy data. Spurious relationships between quasi-global diversity dynamics and quasi-global drivers extends to Red Queen hypotheses of diversity dependence. For diversity dependence to be plausible, organisms or clades must co-occur and bear evidence for their ecological interactions[108], but the former point is rarely considered. Consequently, two clades may superficially display diversity-dependent dynamics at a global scale but could feasibly have been largely geographically separate within the fossil record. Fortunately, our regionalised analyses alleviate this concern by ensuring that geographic context is appropriately controlled prior to modelling of diversity-dependent effects; large diversity fluctuations in the WIS for example will have no bearing on the rates and processes inferred in South Africa.

Taxonomic predictors of extinction risk also showed spatial variability. In contrast to time-continuous biotic or abiotic drivers of diversification measured from geological proxies, taxonomic identities are discrete, intrinsic properties of their fossil occurrences. As taxonomic identity is independent from measurement location, we do not expect spatial sampling biases to impact the taxonomic identities assigned to fossil occurrences. Nonetheless, sampling of fossil occurrences remains spatially biased and so the sampling of taxonomic identities within the fossil record may still be biased in the same way, even if the identities themselves are unaffected. It is therefore conceivable that while a group might show a net pattern of decline in its total fossil record, such a pattern might primarily reflect the dynamics of the group in one particularly well-sampled region, even if that same group flourished in other parts of the globe or was at least not subject to the same degree of extinction risk. At the global level, suborder predictors of genus extinction risk most closely resembled the patterns from Tethys and the East Pacific while the Antarctic and the Atlantic and Gulf were more divergent, but similarities between global and regional extinction risk also changed at the species level. Under the assumption that ammonoid suborder classifications are reasonable proxies for distinct ecomorphological groups, this highlights a complex relationship between discrete ecological traits and extinction risk that may vary with geographic location and taxonomic level. The traditional suborders Phylloceratina and Lytoceratina have long been noted by ammonoid workers for their evolutionary stability compared to other clades[109,110], and it is notable that they appear to possess the lowest extinction risk in our analyses (Fig. 5). By contrast, consistently elevated extinction risk exhibited by the Perisphinctina is consistent with the much higher evolutionary turnover seen in most Late Cretaceous ammonoids[111,112].

While our regionalised approach avoids analysis of an occurrence dataset as though it were a representative sample of the true global trend, it is unlikely to entirely eliminate all sources of bias. Simulation-based studies have demonstrated that PyRate robustly estimates extinction and speciation rates even when there is strong preservation rate heterogeneity through geological time, including for intervals without any sampling (thus modelling the effects of unconformities in the stratigraphic record[34–36]). Preservation rates, however, only consider the numbers of fossils per lineage through geological time[35], which does not necessarily reflect how sampling of different facies may vary at regional scales. As such, birth-death dynamics could still be biased if regional stratigraphic frameworks inconsistently sample different facies through time. The ammonoid fossil record is potentially vulnerable to sea level change driving within-region shifts in sampling of onshore versus offshore depositional environments, although their frequent preservation across a wide range of marine facies[113,114] suggests that their record may still provide a reliable diversification signal even though many taxa tended to inhabit relatively offshore environments. Understanding how facies may affect regional preservation rates is potentially challenging to investigate due to the lack of global stratigraphic databases, but may be achievable at least for the WIS, Atlantic and Gulf, and East Pacific using Macrostrat[115]. Besides stratigraphic sampling issues, the deficit between species and genus richness in some intervals demonstrate that regional records remain prone to taxonomic sampling artefacts. Previous applications of PyRate at both genus and species levels have not reported this effect to our knowledge, but its presence in a fossil record as well-sampled as that of ammonoids suggests that is it likely an issue for other clades as well. Consequently, incomplete sampling of diversity may affect accurate inference of its drivers between taxonomic levels, although the similarity between ammonoid diversity curves at both levels suggests that relative changes in species diversity remain informative in this instance.

In conclusion, while our modelled diversity curves show that ammonoids were affected by global environmental changes during the Late Cretaceous, including short-term responses to OAE2 and long-term correlations with sea level, those curves do not support the hypothesis of long-term global diversity decline[73,76,77] prior to their final extinction at the K-Pg boundary. This conclusion may only be reached when the spatial variability of their diversification processes is properly considered, thus avoiding the pitfalls posed by spatially biased sampling. Instead, their extinction remains attributable to a combination of their planktonic larval ecology and reliance on plankton for food[116], high metabolic rate compared to other cephalopods[117], and potentially narrow geographic ranges of many common taxa[79], rendering them susceptible to the catastrophic collapse of pelagic ecosystems following the Chicxulub asteroid impact at the end of the Cretaceous[73,76,118]. Our regionalised approach also elucidates how the spatial distribution of fossil occurrences may bias our view of 'global' drivers towards the processes responsible for diversity trends in well-sampled regions. The strong signals imposed by Tethys and WIS on their total macroevolutionary history indicate that further work is required to precisely characterise the record of diversity changes in ammonoids, along with the biogeographic nuances of their underlying drivers. Our regionalised approach goes some way towards addressing the issues of spatial heterogeneity and so the ammonoid diversity patterns we present here may still be considered robust, even if their drivers cannot be inferred as confidently, while our taxonomic results demonstrate the potential limitations of proscribing the macroevolutionary fates of clades or ecotypes from spatially biased data. Even so, future work should focus on characterising the regional heterogeneity of diversification processes in other clades to clarify the role of taxonomic affiliation in sensitivity to spatial variation in drivers, as well as how facies variation within regional stratigraphic systems might affect inference of birth-death processes from fossil occurrence data.

Comparison of regional diversification rates to regional drivers will be a crucial step towards overcoming the biases presented by the geological record, although this will not be possible in many instances due to geological and anthropogenic sampling limitations resulting in the lack of one or both components. Nonetheless, regionally variable diversity trajectories may still be expected to show heterogeneous responses to changes in the global average for a candidate driver, itself a mosaic of regionally varied states. Crucially our work does not deny

the existence of global drivers of diversity. For example, the inter-linked effects of temperature and primary productivity have repeatedly emerged as good predictors of species richness in neontological and palaeontological contexts using both empirical and simulation-based approaches[19,25,119,120]. We argue, however, that it is challenging to confidently identify such relationships in the empirical fossil record due to the difficulty of separating spatially biased sampling from genuine spatial variation in drivers of diversity, particularly if a clade's specific ecological responses to a driver strongly differed to the taxon-averaged responses displayed by entire ecosystems. At best, a global driver and global diversity curve may both be reliable and display a significant relationship, but this relationship may be more complex when regional heterogeneity of diversification is considered. At worst, both data sources may be spatially biased, leading to spurious conclusions at regarding the drivers of diversity at broader spatial scales, whilst simultaneously masking the nuances of diversification at more local scales. Coupling of Earth-system model estimates of environmental drivers with diversification models offers a potential solution to this challenging issue, both for speciation patterns[19] and extinction mechanisms like temperature-dependent hypoxia[121,122].

## Methods

### Ammonite occurrence data compilation

We compiled a database of Late Cretaceous ammonite occurrences from several sources: the Palaeobiology Database (PBDB), a published occurrence dataset from Yaccobucci[80], and three unpublished datasets compiled by the authors. This includes vast new occurrence data made accessible through large-scale museum digitisation initiatives[123], alongside occurrences compiled from literature sources not present in the PBDB. To avoid edge effects in downstream analyses, we additionally included all ammonoid occurrences in the PBDB for the preceding Albian stage of the Early Cretaceous. All dataset compilation and revisions were conducted in R (v 4.2.2)[124]. All Cretaceous cephalopod occurrences were downloaded via the PBDB application programming interface on 31/05/22 to avoid potential exclusion of ammonite occurrences misclassified within other cephalopod orders or with poor stratigraphic constraints. Occurrence chronostratigraphy in each dataset was updated according to the Geologic Timescale 2020 (GTS2020)[125] using the `chrono_scale()` function from the fossil-brush R package[126], then palaeocoordinates calculated from their midpoint ages under the PALEOMAP plate rotation model[126–128] using the `palaeorotate()` function with default settings from the palaeo-verse R package[129]. A small number of occurrences could not be rotated and were discarded from the dataset. PBDB occurrences all had locality information present, while missing localities in the other datasets were assigned according to unique combinations of their present-day longitude, latitude, and maximum and minimum stratigraphic ages. All four datasets were then combined, incorporating their occurrence and locality IDs, classifications at the species, genus, family, and order levels, maximum and minimum chronostratigraphic ages, and present-day and palaeogeographic longitude-latitude coordinates.

Next, we revised the taxonomy of the Late Cretaceous occurrences in our composite database. Subgenus classifications recorded within the genus column (commonplace within the PBDB) were deleted, leaving only the genus-level assignment. Suspect and outdated genera were manually reassigned, and family- and order-level classifications of all genera exhaustively updated according to the current literature and the taxonomic expertise of the authors. Further inconsistencies arising from spelling variations between genus names and any remaining discrepancies in higher-level taxonomy were flagged using the `check_taxonomy()` function from fossilbrush and manually resolved. Uncertain species entries (e.g., those with cf. or "?" modifiers) were stripped back to genus-level. Occurrences without genus-level assignments, those named from ammonite jaw elements,

and all remaining non-ammonoid occurrences were discarded. Superfamily and suborder classifications were assigned to each occurrence, using the scheme of Yacobucci[80] for the latter. All taxonomic revisions are documented in the electronic supplement.

Finally, we inspected the stratigraphic ranges of the remaining genera to identify and eliminate incorrectly classified occurrences using several methods. Firstly, stratigraphically suspect occurrences were detected by comparing occurrence stratigraphic ages in the composite database to two reference databases of ammonoid stratigraphic ranges: the Sepkoski Compendium[130] and an unpublished compilation constructed with taxonomic expertise by J.D.W. The Sepkoski Compendium with GTS2020 dating was obtained from fossilbrush, while the compilation from J.D.W was updated to the same chronostratigraphic standard as above. Database comparisons were performed using the `flag_ranges()` function from fossilbrush, which identifies occurrences falling outside their reference ranges. Secondly, misclassified or poorly dated occurrences contributing to anomalously long tails (the portions of a genus stratigraphic duration falling outside a given confidence interval in the distribution of its occurrences) were flagged using the `pacmacro_ranges()` function from fossilbrush. Flagging by this latter method was performed using the kernel density and histogram implementations with tail proportions of 30, 35 and 40% (i.e., the % of the stratigraphic duration occupied by the tails) with the default step size of 0.1 Ma for densifying occurrence durations and default tail confidence intervals of 5% (i.e., the lowermost and uppermost portions of a stratigraphic duration assigned to the tails). Flagged occurrences from both methods were then manually inspected to determine their validity, leading us to discard them from the dataset, revise their genus-level classifications, or update their chronostratigraphic ages. Lastly, all remaining occurrences falling outside the Albian to Maastrichtian stages were discarded from the composite database along with all occurrences with stratigraphic age uncertainties exceeding 10 million years. All rejections are documented in the electronic supplement.

To our knowledge, our final database is the largest and most spatiotemporally comprehensive compilation of Late Cretaceous ammonoids to-date, with fully revised and internally consistent taxonomy, containing 19,536 mostly substage-level occurrences (mean and median age uncertainties of 3.5 and 2.5 Ma, respectively) classified at the order, suborder, superfamily, and genus levels, of which 15,149 (77.5%) are additionally classified to species level, and 9962 (50.1%) are occurrences from literature not yet incorporated into publicly accessible biodiversity databases, or true 'dark' data from museum collections. Consequently, it is well-suited for analysis of their diversity dynamics in the interval leading up to their demise at the end-Cretaceous mass extinction.

### Spatial subsampling

To determine how ammonoid diversity dynamics differed between different parts of the globe and to circumvent potential biases introduced by spatially heterogenous sampling in the fossil record, we subsampled our composite dataset into eight biogeographically relevant regions with ammonoid occurrence records covering the full extent of the Albian to Maastrichtian: Tethys (corresponding to Europe and North Africa), West Africa, South Africa, Antarctic (corresponding to the James Ross Basin), West Pacific (corresponding largely to Japan), East Pacific (corresponding to the North American Pacific seaboard), the WIS, and the Atlantic Gulf (corresponding to the North American Atlantic and Gulf Coast). These regions were largely identified by plotting the palaeogeographic distribution of our composite dataset at five million-year intervals through the Late Cretaceous onto the PALEOMAP highstand reconstructions of Scotese and Wright (2018)[128], while the decision was made to separate the otherwise geographically continuous Atlantic and Gulf from the WIS as they constitute different water masses and show marked

endemic differences in their ammonoid faunas[131,132], supporting their designation as distinct bioregions.

We followed the methodology of Flannery-Sutherland et al.[65] to eliminate spatial sampling bias in each region, using the corresponding R functions provided in their ESM. We constructed fixed-size spatial windows to subsample the occurrence data for each region using the `spacetimewind()` function, permitting each window to slide in a constant direction between geological stages to accommodate continental drift of their target biogeographic regions. To remove any remaining effects of spatial sampling heterogeneity, the data in each region was further subsampled within each geological stage to target spatial extent based on minimum spanning tree (MST) length. MST length is correlated with a variety of other spatial properties[12] and so is a robust metric for the spatial extent of a geographically resolved dataset. Subsampling was implemented using the `spacetimestand()` function which spatially bins the occurrence data within a given region and geological interval using a hexagonal grid generated by the `hexagrid()` function from the icosa R package[133], thus avoiding any latitudinal distortions in grid cell area, calculates the MST connecting all occupied grid cell centres, and sums the great circle lengths of each of its segments. The function then progressively discards grid cells forming the tips of the MST, prioritising those with the fewest occurrences, until the closest possible match to the target MST length is achieved. After spatial standardisation, apparent changes in taxonomic diversity within a region may still be driven by temporally heterogeneous sampling. Consequently, a temporally standardised diversity curve should not be driven by any residual fluctuations in spatial extent if spatial standardisation was effective. We therefore calculated sampling-corrected genus diversity at stage level through time in each region by shareholder quorum subsampling (SQS) with a sampling quorum of 0.5, implemented within `spacetimestand()` using the `estimateD()` function from the iNEXT R package[134] (Supplementary Figs. 1–9). SQS is the most robust subsampling method for estimating diversity from fossil occurrence data and so is well-suited to this purpose[135]. We then conducted one-tailed Pearson and Spearman correlations between genus-level SQS diversity, and longitude range, latitude range and MST length for each spatially standardised region to determine if increased spatial extent was associated with increased diversity. To determine how global ammonoid diversity dynamics related to its regional components through the Late Cretaceous, we also spatially standardised the entire composite dataset and tested for correlations between genus-level SQS diversity and spatial extent, following the above protocol (Supplementary Data 1). Collectively, our regions encompass 69.3% of the occurrences in the total dataset, while the standardised global dataset encompasses 94.5%.

## Diversity dynamics

Diversity dynamics were quantified for each spatially standardised regional dataset in a hierarchical Bayesian framework using PyRate (v 3.0)[34–36]. PyRate jointly estimates the preservation rate underlying the occurrence records of an incompletely stratigraphically sampled set of fossil lineages, sampling-corrected times of origination and extinction of those lineages, and the corresponding origination and extinction rates arising from their true stratigraphic durations[35]. PyRate models origination and extinction rates using a birth-death process describing the expected numbers of lineage origination and extinction events per unit of time, and preservation rates using a Poisson process describing the expected number of fossil occurrences per lineage per unit of time, with posterior distributions of the parameters of each process estimated using Markov Chain Monte Carlo (MCMC). PyRate can implement several different Poisson preservation processes, including the potential for their variation between lineages and through time, and uses reverse jump MCMC sampling to estimate the number and timing of rate shifts in the birth-death process, along with their statistical support[36]. By jointly modelling the processes responsible for a set of taxonomically resolved fossil occurrence data, along with their uncertainties, PyRate provides substantially more accurate estimates of origination and extinction rates through geological time compared to traditional methods[36,136] and the method has consequently received widespread use by palaeontologists working on a wide variety of different clades (e.g., ferns[45], dinosaurs[47], marine megafauna[137]).

Input files for PyRate were generated in R from the global and regional datasets at genus and species levels. Each input file contains 10 age-randomised replicates of the original dataset where the age of each occurrence is randomly fixed within the bounds of its stratigraphic uncertainty. This procedure was performed such that occurrences from the same locality were consistently assigned the same randomised age in each replicate, following previous recommendations[138]. The best-fitting preservation model for each dataset was identified by maximum likelihood, using the `-PPmodeltest` command of PyRate) to compare Akaike's information criterion (AIC)[139] scores for a homogenous Poisson process (HPP), an inhomogenous Poisson process (NHPP), and a time-variable Poisson process (TPP) consisting of user-defined rate shifts bounding constant-rate intervals (Supplementary Data 2). For the TPP model, rate shifts (PyRate option `-qShift`) were set as the boundaries between the geological stages of the Late Cretaceous. The TPP model was identified as best-fitting for each regional dataset, except for Antarctica at the genus-level where the HPP model was best-fitting. Initial analysis under the HPP model, however led to unrealistic extension of Antarctic ammonoid genus ranges into the Cenozoic by several million years, despite the empirical fossil record clearly documenting their rapid demise at the end of the Cretaceous. Consequently, we elected to use the TPP model in all cases, as the HPP model was unable to adequately capture this particular case of total loss of preservation rate in the geological past resulting from sudden and complete clade extinction well before the present day. PyRate uses a gamma distribution with shape = 1.5 as the default prior on the vector of bin-wise preservation rates when using the TPP model. We assigned a vague exponential hyper-prior on the rate parameter of the gamma distribution (PyRate option `-pP 1.5 0`) and permitted PyRate to estimate the empirical rates during the analysis, reducing subjectivity in prior selection and allowing the analysis to better adapt to the input data. Each analysis was additionally constrained to only search for rate shifts within the Albian to Maastrichtian to further reduce edge effects induced by the temporal bounds of our data (PyRate option `-edgeshift 113.2 66`).

We ran PyRate on each age-randomised dataset for 100 million generations, sampling every 50,000, using the reverse jump MCMC algorithm to efficiently search for shifts in origination and extinction rates rather than set any such shifts a priori (PyRate options `-A 4`). Log files were analysed in Tracer (v 1.7.2)[140] to determine that stationarity had been achieved. The first 10% of each log file was discarded as burn-in, then each log file combined into a final log using the `-combLog` command of PyRate, enabling the age uncertainties in the original fossil occurrence data to be incorporated into our results. Origination and extinction rates, along with Bayes Factor (BF) support for any rate shifts, and mean times of origination and extinction for each lineage were calculated for each region using the `-plotRJ` and `-ginput` commands of PyRate (Supplementary Figs. 10–18). To assess the statistical significance of rate shifts, we took logBF > 2 and logBF > 6 as positive and strong support respectively[141]. Range-through diversity was calculated from the preservation-corrected lineage stratigraphic durations at 0.5 Ma intervals in R.

## Drivers of origination and extinction

We used a multivariate birth-death (MBD) model to identify potential drivers of ammonoid diversification dynamics, implemented in PyRateMBD[45]. Origination and extinction rates within the birth-death process are modelled as linear or exponential correlations of time-continuous variables. Rates are derived from the preservation-

corrected lineage durations estimated by a previous PyRate analysis, circumventing the need to account for sampling heterogeneity within the MBD model itself. The strength and sign of the correlation functions are jointly estimated for each variable using MCMC, with a horseshoe prior on each variable controlling for over-parameterisation and the potential effects of multiple testing by shrinking each correlation parameter around zero. Significant correlations consequently show shrinkage weights which differ substantially from zero ($W > 0.5$)[45].

For biotic influences, we considered the effects of ammonoid diversity on their own origination and extinction rates (diversity dependence), the influence of nutrient availability and the pressure of pelagic predators/competitors. Ammonoid diversity was calculated directly within PyRateMBD from the preservation-corrected lineage durations used by the analysis. The diversities of calcareous nannoplankton and planktic foraminifera[142] were used as proxies for particulate nutrient availability in the water column. Ichthyolith accumulation rates[143,144] was used as a proxy for the abundance of large pelagic fish which were the most likely predators or competitors of ammonites[145,146], as opposed to larger apex predators like mosasaurs. For abiotic influences, we considered long-term sea level[106], atmospheric $CO_2$[147], and sea surface temperature[148]. All variables were taken from the electronic supplementary materials of each source, linearly interpolated at 0.1 Ma intervals, then rescaled to the range [0,1] to avoid biases induced by differences in their absolute magnitudes.

We ran PyRateMBD on the 10 sets of lineage origination and extinction times estimated in each regional genus- and species-level PyRate analysis for 50 million generations, sampling every 50,000, using exponential then linear correlation functions. To eliminate the potential impacts of total and near instantaneous lineage extinction at the end of the Cretaceous from affecting our results, all MBD analyses were temporally restricted to 100.5–68 Ma. Log files were analysed in Tracer to check for stationarity, then combined log files produced as above. We then compared the fits of the exponential versus the linear MBD models for each dataset in R using AIC for MCMC samples[149] and Bayes factors[141] calculated from the harmonic means of each model likelihood. In almost all cases we found that the fits of these equally complex models could not be readily distinguished from one another ($\Delta_{AIC} < 2$, logBF < 2; Supplementary Data 3), aside from the genus-level analyses for the Antarctic where the exponential model was favoured (logBF = 5.31). Regardless, the statistical significance of virtually all the correlation parameters were unchanged by the choice of correlation function, aside for the genus-level analyses for the West Pacific where the exponential model was marginally better fitting ($\Delta_{AIC} = 0.05$). Consequently, we interpreted results from the exponential models in all cases (Supplementary Figs. 19–27, Supplementary Data 4–12), although the results from linear models are also available in the electronic supplement (Supplementary Figs. 28–36, Supplementary Data 4–12).

### Ecological correlates of extinction
In addition to examining the influence of temporally varying abiotic and biotic drivers on ammonoid diversification, we additionally tested for discrete ecological correlates of their extinction risk using a multi-trait-dependent extinction (MTE) model, implemented in PyRateMTE[150]. Likelihood of lineage durations within the birth-death process is modelled as a function of discrete, trait-based multipliers and a mean extinction rate. As with the MBD model, lineage durations are derived from the preservation-corrected speciation and extinction times estimated in a previous PyRate analysis. The trait multipliers enable lineage-wise deviations from the mean extinction rate, with mean extinction rate and multiplier strengths estimated using MCMC. Model priors and hyperpriors control for over-parameterisation by shrinking the multipliers towards equal values (i.e., no differential effects on extinction rate), along with Bayesian variable selection,

which places 95% of the prior probability on the absence of an effect for each trait then removes traits deemed to have no substantial effect on rate variation. Traits selected as having an effect, which additionally show posterior frequencies greater than 51.4% (equivalent to logBF > 6) are taken as statistically significant.

Suitable ecological traits may be derived directly from categorical aspects of an organism's life mode (e.g., filter feeder, deposit feeder, active hunter etc.) or from discretisation of a continuous trait (e.g., occupied depth range). Many of traits are difficult to directly characterise for ammonoids, but their higher taxonomic classifications are considered to map onto ecologically distinct life modes. Consequently, we used the suborder and superfamily classifications (five and twelve categories respectively) as ecological proxies of extinction risk in the MTE model, under the assumption that these classifications subsume sets of clade diagnostic characters that are also responsible for their distinct ecological modes. We ran PyRateMTE on the 10 sets of lineage origination and extinction times estimated in each regional genus- and species-level PyRate analysis for 10 million generations, sampling every 10,000. To eliminate the potential impacts of total and near instantaneous lineage extinction at the end of the Cretaceous from affecting our results, all MBD analyses were temporally restricted from 100.5 Ma to 68 Ma. Log files were analysed in Tracer to check for stationarity, combined log files produced as above, then each model parameter summarised as their means (Supplementary Data 13–14).

### Reporting summary
Further information on research design is available in the Nature Portfolio Reporting Summary linked to this article.

## Data availability
All code and data generated in this study, including source data for main figures and Supplementary Figs. and Tables, has been deposited in the FigShare database at: https://doi.org/10.6084/m9.figshare.25563633.

## Code availability
PyRate, PyRateMBD and PyRateMTE are freely available on Github (https://github.com/dsilvestro/PyRate). All scripts used to conduct our analyses are available in the electronic supplement for this paper at: https://doi.org/10.6084/m9.figshare.25563633.

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

## Acknowledgements

J.F.S. was funded by NERC GW4 + DTP studentship S100065-138/123. The compilation of ammonite occurrence data from the Eastern Pacific was made possible through funding to A.J.W.H. from the National Science Foundation (NSF 1561429, NSF 1902262). This work arose from a thesis submitted for partial fulfilment of an MSc degree by C.C. at the University of Bristol.

## Author contributions

A.J.W.H., C.E.M. and J.D.W. contributed ammonite occurrence datasets. J.F.S. C.C. and J.D.W. compiled the composite occurrence dataset. J.F.S and C.C. conducted the spatial standardisation and diversification rate analyses. J.F.S. conducted the diversification driver analyses. J.F.S., C.C. and J.D.W. wrote the manuscript with input and comments from A.J.W.H., C.E.M. and N.L.

## Competing interests

The authors declare no competing interests.
