## [Peer Review File · Nature Communications]

Late Cretaceous ammonoids show that drivers of diversification are regionally heterogeneousReviewers' Comments:

Reviewer #1:

Remarks to the Author:

I was really enjoyed to read this manuscript!

The driver of diversification is one of the most attractive topics in both paleontologists and biologists. The authors adopted the state-of-the-art Bayesian method, PyRate, to recovery "global" and regional diversity dynamics of Late Cretaceous ammonoids, and found strong heterogeneity in diversity dynamics among different regions and highly variable interactions between diversity and different biotic and abiotic factors in variable tempo-spatial scales. The manuscript is well written, and the figures are excellent.

I have only a few concerns/comments as follow:

I still can't accept that species richness can lower than genus richness in some intervals in some regions. This is not true based on our natural classification system. The authors could treat, for instance "Haploceras sp.", as a species to overcome this problem.

Lines. 144-169: It's a bit hard to read here. Why don't you describe the diversity dynamics in the eight regions in one paragraph. For instance: "Declines in genus and species richness occurred during the interval containing OAE2 in all four," At first glance, I was confused, what are the four regions? I can only find the answer (East Pacific, Tethys, WIS and Atlantic and Gulf) when I read the whole paragraph.

Line 353. "Middle" => "Early".

Line 577-578: Do you mean five superfamily and twelve suborders? Please check.

Reviewer #2:

Remarks to the Author:

Spatiotemporal variations of biodiversity impact our understanding of past evolution. This manuscript focuses on the debate of what drives diversification and whether we can faithfully identify these drivers from the fossil record.

I do not have major concerns on the technical part of the paper. However, the author(s) soundly criticized past studies and, in some sentences, all palaeontologists. I am not sure if this is the best solution to promote this paper, but this is up to the authors. From an absolutely objective perspective, most researchers need a simple solution for many applications of biodiversity changes e.g., in palaeoenvironment studies. The approach provided by the author(s), though impressive and better, is not simple, nor a solution to the general problem.

Nevertheless, this is a well-written manuscript that will add new insights into our understanding of drivers of past extinction, recovery, and diversification.

Line 69-71: True, but the words are rather harsh.

Line 137: WIS first occurred here. But the abbreviation was only explained in line 421.

Lines 144-155: When reading this paragraph, the question popped up: Would the bias of sampling efforts completely rule out?

Lines 242-243: the sentence "regression-driven loss of habitat space is more commonly associated with elevated extinction rates" needs references.

Lines 248-250: difficult to follow, explain why prevalence in the smaller sampling regions and absence in the larger occur.

Lines 250-252: regional ammonoid origination and extinction rates showed marked heterogeneity. How much difference can be inferred as marked heterogeneity? Please explain.

Lines 288-293: Not very clear here. Taxonomic assignments are not affected by measurement location. This seems to contradict the idea of spatial heterogeneity mentioned earlier. And the following sentence "fossil occurrences associated with taxonomic assignments display geographically biased sampling" doesn't seem to make sense. Explain in more detail and give some references.
Lines 309-312: What then was the reason (driver) for ammonoid extinction?
Line 311-312: the sentence "the hypothesis of long-term global diversity decline prior to their final extinction at the K-Pg boundary" needs references.

Reviewer #3:

Remarks to the Author:

I laude the authors for their focus on regional diversity trends as a way to understand "global" diversification patterns of ammonoids during the Late Cretaceous. As Geerat Vermeij stated in 2003, global diversity does not really mean what it implies given the well-established biases in the known fossil record at the global scale. This study is an interesting and important contribution to the understanding of a major group of animals as they approached their ultimate demise. The results indicate that despite parsing out global data into distinct regions, ammonoid diversity was not waning during the Late Cretaceous as many workers have suggested. Moreover, the results suggest that drivers of regional diversity patterns are more likely driven by regional drivers, such as temperature and sea-level change.

Overall, this study highlights an exciting approach for assessing the evolutionary history of Late Cretaceous ammonoids and potential application to other major invertebrate groups in the fossil record. The methods also seem sound to me, particularly in their approach for controlling sampling biases. I only have a few minor comments to make regarding the discussion and methodology, otherwise I believe this manuscript should be accepted once these issues have been addressed.

Minor comments:

Sea-level change has long been attributed to be a driver of both extinction and origination in dynamics in the fossil record. However, evidence suggests that sea-level change itself does not meaningfully effect macroevolutionary dynamics, regardless of spatial scope (Holland 2012, Paleobiology). Given that ammonoids are pelagic animals and even less likely to be affected by sea-level change, I fail to see the reason to include sea-level analysis in this study. Sea-level change will have a stronger relationship with the types of facies preserved, with relatively deeper-water facies being deposited during times of sea level rise, which are more likely to preserve ammonoid fossils than shallower water facies. Which leads into my next comment.

Again, while I appreciate the effort of the authors to consider the regional heterogeneity of diversity dynamics, there is no mention regarding the nature of the stratigraphic record at the regional scale. While ammonoids are preserved across the majority of marine facies, they are most likely to be live and preserve in relatively deeper-water environments (80–400m; Ritterbush 2014, J. of Zoology; Lukeneder et al. 2010, Earth Planet. Sci. Lett.). As such, the structure of the stratigraphic record regionally would have a greater effect on diversity rather than sampling biases. I wonder if stratigraphic packages regionally would potentially have a 1st order control on diversity. I do realize, however, the difficulty in actually testing this, given the poor state of facies/deposition environment data in the PBDB and lack of global stratigraphic packages in Macrostrat. Just a thought.

Lastly, while sea-surface temperatures might have mixed effects on diversity in this study, it is possible that temperature in conjunction with oxygen might be a better environmental component to assess. While modern cephalopods are sensitive to temperature, they have behavioral adaptations to compensate for increased temperatures, but not for temperature-dependent hypoxia.

Otherwise, great and interesting study!

Response to Reviewers

Reviewer comments are given in regular text, our responses are in red.

Reviewer #1:

I was really enjoyed to read this manuscript! The driver of diversification is one of the most attractive topics in both paleontologists and biologists. The authors adopted the state-of-the-art Bayesian method, PyRate, to recovery “global” and regional diversity dynamics of Late Cretaceous ammonoids and found strong heterogeneity in diversity dynamics among different regions and highly variable interactions between diversity and different biotic and abiotic factors in variable tempo-spatial scales. The manuscript is well written, and the figures are excellent. I have only a few concerns/comments as follow:

We are delighted by the reviewer’s positive response to our work and thank them for their endorsement.

I still can’t accept that species richness can lower than genus richness in some intervals in some regions. This is not true based on our natural classification system. The authors could treat, for instance “Haploceras sp.”, as a species to overcome this problem.

We agree that genus richness exceeding species richness is problematic under the nature of systematics, but we do not posit that it is a true pattern. Instead we recognise it as an artefact of empirical palaeontological occurrence datasets that results from a deficit between occurrences identifiable at species level versus genus level. That this artefact is present in a clade as well sampled as ammonoids suggests that it is likely present elsewhere in the fossil record. The reality of incomplete sampling effects between taxonomic levels is an ugly one, but we assert that this is an important limitation of empirical palaeontological occurrence data which should be presented honestly, given the valid methodological concerns it raises.

We do not believe that adding placeholder species to a formal PyRate analysis would be a justifiable approach for three reasons:

1. PyRate models preservation rates to help estimate true lineage durations, so the taxonomic resolution of an occurrence will affect how it contributes to genus- versus species-level preservation rates. Dragging genus level occurrences to the species level would imprint genus-level rates on the modelled species-level rates, distorting estimated species durations and their origination-extinction dynamics.
2. While the purpose of placeholder species would be to meet the required richness ratio of at least one species per genus, it runs the risk of lumping genuinely separate (albeit unidentified) species into one composite record without any way of detecting this distortion.
3. The logic behind assigning all genus-level occurrences of a taxon to one placeholder species means that that all occurrences which contribute information on the genus-level duration of that taxon should be treated in the same way. This would necessitate inclusion of species-level occurrences within their genus-wise placeholder species, despite already representing their own lineages, and resulting in an analysis which is then largely collapsed back to the genus level. Conversely, inclusion of solely genus-level occurrences in placeholder taxa would be an arbitrary choice to make species-level dynamics reflect genus level dynamics, even though they are ultimately different signals.

A simpler, post hoc solution is for the reader to interpret the genus level curve as a lower-bound estimate on true species richness in instances where species richness falls below genus richness. However, the high correlation values between genus and species richness in each region suggest that relative changes in the later still faithfully reflect changes in the former, lending support to admittedly imperfect signal of species diversification we recover

here. Ultimately, we believe that it is better to present both sets of curves as they stand, given there is still no robust linkage in the literature between genus and species-level dynamics, and acknowledge their imperfections clearly in the discussion. To aid in this, we have made explicit reference to the issue in the caption for fig. 3 (L1056–1058) and direct the reader to the longer discussion of this issue in the text (L340–347).

Lines. 144-169: It's a bit hard to read here. Why don't you describe the diversity dynamics in the eight regions in one paragraph. For instance: "Declines in genus and species richness occurred during the interval containing OAE2 in all four," At first glance, I was confused, what are the four regions? I can only find the answer (East Pacific, Tethys, WIS and Atlantic and Gulf) when I read the whole paragraph.

We agree that our original phrasing did not aid with manuscript clarity here. We now describe the dynamics in one paragraph as the reviewer suggests (L144-169). Our original structure was designed to highlight how regions away from the Global North depart from the 'global' diversity signal, so we have tried to retain this in the revised structure as we view it as a key result in our study.

Line 353. "Middle" => "Early".

Change made (L401)

Line 577-578: Do you mean five superfamily and twelve suborders? Please check.

Quite right, these were misordered, change made (L625)

I tried several times and can't access the code via the link provided above.

We apologise for this issue. We have revised our ESM in light of this review and have made this new version available via the manuscript submission portal using Nature's FigShare service. We hope that this has addressed the problem, but if not then please get the editor to contact us and we will endeavour to make the full ESM available in another way

Reviewer #2:

Spatiotemporal variations of biodiversity impact our understanding of past evolution. This manuscript focuses on the debate of what drives diversification and whether we can faithfully identify these drivers from the fossil record. I do not have major concerns on the technical part of the paper. However, the author(s) soundly criticized past studies and, in some sentences, all palaeontologists. I am not sure if this is the best solution to promote this paper, but this is up to the authors.

We are pleased that the reviewer views our paper as technically sound. We agree that our original phrasing could be interpreted as a little damning of all palaeontologists, which is certainly not our intention. We have modified our phrasing to reduce this possibility. We instead view the need for spatially sensitive analysis as the 'promoting' point for the work.

From an absolutely objective perspective, most researchers need a simple solution for many applications of biodiversity changes e.g., in palaeoenvironment studies. The approach provided by the author(s), though impressive and better, is not simple, nor a solution to the general problem. Nevertheless, this is a well-written manuscript that will add new insights into our understanding of drivers of past extinction, recovery, and diversification.

We thank the reviewer for their positive perspective on the contributions of our manuscript to the field. We agree that our analytical strategy is not the most straightforward to implement, but this is largely because we aimed to analyse rates rather than standing diversity. Measuring the latter in a spatially sensitive framework is much easier to achieve using standard analytical tools and R packages and is a more achievable solution for future workers. We view our take home message as the need for spatially sensitive analysis to effectively tackle questions of extinction, recovery and diversification in the fossil record.

Line 69-71: True, but the words are rather harsh.

We have revised our phrasing to 'some palaeontologists' (L69). This is a more accurate reflection of the state of the field and implicitly acknowledges the excellent efforts of other palaeontologists to move away from global analyses towards spatially sensitive frameworks.

Line 137: WIS first occurred here. But the abbreviation was only explained in line 421.

We have ensured that the abbreviation is now provided at the first instance of the complete phase (L95).

Lines 144-155: When reading this paragraph, the question popped up: Would the bias of sampling efforts completely rule out?

Our analytical strategy is designed to minimise various sources of sampling bias as much as possible, but we agree that it is probably impossible to eliminate its effects entirely. We do not address this point within the lines referenced by the reviewer to keep the results free from any discussion but have referred to the issue of residual sampling bias as a new section later in the manuscript (L324–340). We believe that this inclusion greatly improves the quality of our work and thank the reviewer for suggesting that we consider the issue of sampling in more detail

Lines 242-243: the sentence “regression-driven loss of habitat space is more commonly associated with elevated extinction rates” needs references.

We have added references to support the impact of regression on extinction, but we have also caveated that this relationship is not necessarily consistent across space, which helps to clarify why rising sea levels might instead coincide with elevated extinction rates (L243–250).

Lines 248-250: difficult to follow, explain why prevalence in the smaller sampling regions and absence in the larger occur.

This was a remark on the prevalence of significant correlations between sp/ex and ammonoid diversity in the smaller sampling regions, so we have added a reference to the figure and reworked the phrasing for clarity. We have also added a comment with references on a possible explanation for this finding based on the literature – that RQ effects are overwhelmed by CJ effects at larger spatial scales (L260–264).

Lines 250-252: regional ammonoid origination and extinction rates showed marked heterogeneity. How much difference can be inferred as marked heterogeneity? Please explain.

We have removed 'marked' as this phrasing was vague with no clear basis for determining if difference is marked or otherwise (L265).

Lines 288-293: Not very clear here. Taxonomic assignments are not affected by measurement location

We were a little uncertain of whether the reviewer is giving their interpretation of what we said in the manuscript in relation to the next point raised, or simply stating the above point plainly. We aimed to make the same point in the manuscript (L304–305), so we hope that we have understood the reviewer here and that our response below addresses their intended concern.

Lines 288-293 (cont): This seems to contradict the idea of spatial heterogeneity mentioned earlier. And the following sentence “fossil occurrences associated with taxonomic assignments display geographically biased sampling” doesn’t seem to make sense. Explain in more detail and give some references.

We disagree that independence of taxonomic assignment from spatial location contradicts the idea of spatial heterogeneity. Our point is that the sampling of fossils as data points is spatially biased, but that this spatial bias will not have any impact on the taxonomic identities assigned to those data points (L304–309). *Baculites* sp. identified in one location will belong to the same taxonomic unit as *Baculites* sp. from another location, even if one of those locations is disproportionately better sampled than the other. We have revised our phrasing to help ensure that this point is clear.

Lines 309-312: What then was the reason (driver) for ammonoid extinction?

Sentence and references added for the current stance in the literature on ammonoid extinction at the K-Pg (L354–358). We did this at a different point in the manuscript to the lines where the reviewer raises their comment to keep our conclusions as free as possible from inferences made elsewhere in the literature.

Line 311-312: the sentence “the hypothesis of long-term global diversity decline prior to their final extinction at the K-Pg boundary” needs references.

References added (L351).

Reviewer #3:

I laude the authors for their focus on regional diversity trends as a way to understand “global” diversification patterns of ammonoids during the Late Cretaceous. As Geerat Vermeij stated in 2003, global diversity does not really mean what it implies given the well-established biases in the known fossil record at the global scale. This study is an interesting and important contribution to the understanding of a major group of animals as they approached their ultimate demise. The results indicate that despite parsing out global data into distinct regions, ammonoid diversity was not waning during the Late Cretaceous as many workers have suggested. Moreover, the results suggest that drivers of regional diversity patterns are more likely driven by regional drivers, such as temperature and sea-level change.

We are pleased by reviewer’s positive response on our work and thank them for drawing our attention to Vermeij (2003), whose point of whether global trends are actually meaningful is very pertinent to our work, so we have added this reference to the introduction (L65).

Overall, this study highlights an exciting approach for assessing the evolutionary history of Late Cretaceous ammonoids and potential application to other major invertebrate groups in the fossil record. The methods also seem sound to me, particularly in their approach for controlling sampling biases. I only have a few minor comments to make regarding the discussion and methodology, otherwise I believe this manuscript should be accepted once these issues have been addressed.

We are pleased that the reviewer considers our approach to be robust and thank them for their additional comments, which we found to be very thought provoking.

Minor comments:

Sea-level change has long been attributed to be a driver of both extinction and origination in dynamics in the fossil record. However, evidence suggests that sea-level change itself does not meaningfully effect macroevolutionary dynamics, regardless of spatial scope (Holland 2012, Paleobiology). Given that ammonoids are pelagic animals and even less likely to be affected by sea-level change, I fail to see the reason to include sea-level analysis in this study. Sea-level change will have a stronger relationship with the types of facies preserved, with relatively deeper-water facies being deposited during times of sea level rise, which are more likely to preserve ammonoid fossils than shallower water facies. Which leads into my next comment.

We included sea level due to its effect on the extent of shallow marine shelf area. While ammonites are pelagic, we nonetheless expect increased volume of pelagic habitat overlying productive shallow marine shelf area to promote their diversification. Additionally, many heteromorph ammonitez, which became increasingly diverse through the Late Cretaceous and are well represented in our dataset, were nektobenthic rather than pelagic and so shallow marine shelf area will be even more relevant to these taxa We chose to use sea level as a proxy rather than a direct measurement of shelf extent due to ongoing uncertainty between palaeogeographic reconstructions, which would only be compounded by the uncertainty in sea level change itself.

Holland (2012) did find that that the relationship between sea level and shallow marine area was complex and asserted that at local scales sea level will not be a simple predictor for diversity under the species area effect. Holland (2012) also showed, however, that at regional scales that sea level and shallow marine area remain positively correlated (Fig. 4 in paper). Additionally, Holland (2012) did not actually conduct any formal analyses of empirical diversity versus changing area under sea level and simply predicted diversity as a function of benthic area using the Arrhenius equation. More recent studies of the empirical fossil record instead provide support for long term sea level as having predictive power at larger spatial scales (e.g., Roberts and Mannion 2019, *loc cit*).

Ultimately, we assert that sea level remains a worthwhile predictor to test, but we accept that its relationship with diversity may be complex. We have added new material to the discussion to properly address this possibility (L241–250). At best, sea level had a genuine effect and is captured by our analyses. At worst, its effects could be ignored as excluding it from the PyRateMBD analyses would not impact the modelled effects of the other predictor variables.

Again, while I appreciate the effort of the authors to consider the regional heterogeneity of diversity dynamics, there is no mention regarding the nature of the stratigraphic record at the regional scale. While ammonoids are preserved across the majority of marine facies, they are most likely to be live and preserve in relatively deeper-water environments (80–400m; Ritterbush 2014, J. of Zoology; Lukeneder et al. 2010, Earth Planet. Sci. Lett.). As such, the structure of the stratigraphic record regionally would have a greater effect on diversity rather

than sampling biases. I wonder if stratigraphic packages regionally would potentially have a 1st order control on diversity. I do realize, however, the difficulty in actually testing this, given the poor state of facies/deposition environment data in the PBDB and lack of global stratigraphic packages in Macrostrat. Just a thought.

It has been previously determined that PyRate provides robust results even when confronted by strong preservation rate heterogeneity. As such, our regionalised results will account for preservation rate variation induced by regional stratigraphic architecture. We still agree, however, that there will be a trade off between biological signal versus stratigraphic noise in relation to region size. As the reviewer points out, this would be challenging without proper stratigraphic data for each region, but it is still worth pointing this out in the manuscript. No work has been done on how facies architecture might bias PyRate analyses and this will certainly be important to tackle in future work, particularly given the reviewer's pertinent point regarding sea level controlling facies types. We have added a paragraph to the discussion to outline these issues and believe that it significantly improves the presentation of our study limitations (L324–340).

Lastly, while sea-surface temperatures might have mixed effects on diversity in this study, it is possible that temperature in conjunction with oxygen might be a better environmental component to assess. While modern cephalopods are sensitive to temperature, they have behavioural adaptations to compensate for increased temperatures, but not for temperature-dependent hypoxia.

As with the previous issue of the lack of stratigraphic databases to test for facies-driven biases, our temperature and redox data are unfortunately insufficient to properly tackle this issue as they are one dimensional time series which summarise global environmental states, rather than providing the necessary information on how temperature and hypoxia spatially co-varied between regions. A couple of studies have convincingly demonstrated this effect in the fossil record, but all have spatially resolved estimates of oxygen availability and temperature from earth system models coupled with metabolic-limitation models of marine invertebrates. Such analyses fall beyond the scope of this study, but they also provide excellent examples of spatially explicit approaches to modelling diversification which can additionally solve the issues we raise regarding spatially biased proxy records. We have added reference to these model-based studies in our discussion to provide a clear pathway for future work to pursue (L390–392), as well as using the empirical point of temperature-dependent hypoxia to present the potential limitations of our study more thoroughly (L251–258), although recent meta-analysis of cephalopod temperature tolerances found consistently negative effects of warming across lineages and ontogenies (Borges et al., 2023, *loc cit*).

Otherwise, great and interesting study!

Thanks again!

Reviewers' Comments:

Reviewer #1:

Remarks to the Author:

I appreciated the authors' effort to address my comments. Looking forward to see the publish of this intriguing paper!

Reviewer #2:

Remarks to the Author:

Dear authors and editor,

Thanks for revising the manuscript.

I have another careful read of this paper and believe it is ready for publication.

Best regards,

Yadong Sun

Reviewer #3:

Remarks to the Author:

I originally reviewed this manuscript. The authors responded and addressed all my comments and have substantially improved the manuscript. The authors responses were thoughtful to not only my comments but also those made by the other reviewers as well. With the improvement and clarifications made in the new submission, I believe this manuscript warrants publication in Nature Communications.

Response to Reviewers

Reviewer comments are given in regular text, our responses are in red.

Reviewer #1:

I was really enjoyed to read this manuscript! The driver of diversification is one of the most attractive topics in both paleontologists and biologists. The authors adopted the state-of-the-art Bayesian method, PyRate, to recovery “global” and regional diversity dynamics of Late Cretaceous ammonoids and found strong heterogeneity in diversity dynamics among different regions and highly variable interactions between diversity and different biotic and abiotic factors in variable tempo-spatial scales. The manuscript is well written, and the figures are excellent. I have only a few concerns/comments as follow:

We are delighted by the reviewer’s positive response to our work and thank them for their endorsement.

I still can’t accept that species richness can lower than genus richness in some intervals in some regions. This is not true based on our natural classification system. The authors could treat, for instance “Haploceras sp.”, as a species to overcome this problem.

We agree that genus richness exceeding species richness is problematic under the nature of systematics, but we do not posit that it is a true pattern. Instead we recognise it as an artefact of empirical palaeontological occurrence datasets that results from a deficit between occurrences identifiable at species level versus genus level. That this artefact is present in a clade as well sampled as ammonoids suggests that it is likely present elsewhere in the fossil record. The reality of incomplete sampling effects between taxonomic levels is an ugly one, but we assert that this is an important limitation of empirical palaeontological occurrence data which should be presented honestly, given the valid methodological concerns it raises.

We do not believe that adding placeholder species to a formal PyRate analysis would be a justifiable approach for three reasons:

1. PyRate models preservation rates to help estimate true lineage durations, so the taxonomic resolution of an occurrence will affect how it contributes to genus- versus species-level preservation rates. Dragging genus level occurrences to the species level would imprint genus-level rates on the modelled species-level rates, distorting estimated species durations and their origination-extinction dynamics.
2. While the purpose of placeholder species would be to meet the required richness ratio of at least one species per genus, it runs the risk of lumping genuinely separate (albeit unidentified) species into one composite record without any way of detecting this distortion.
3. The logic behind assigning all genus-level occurrences of a taxon to one placeholder species means that that all occurrences which contribute information on the genus-level duration of that taxon should be treated in the same way. This would necessitate inclusion of species-level occurrences within their genus-wise placeholder species, despite already representing their own lineages, and resulting in an analysis which is then largely collapsed back to the genus level. Conversely, inclusion of solely genus-level occurrences in placeholder taxa would be an arbitrary choice to make species-level dynamics reflect genus level dynamics, even though they are ultimately different signals.

A simpler, post hoc solution is for the reader to interpret the genus level curve as a lower-bound estimate on true species richness in instances where species richness falls below genus richness. However, the high correlation values between genus and species richness in each region suggest that relative changes in the later still faithfully reflect changes in the former, lending support to admittedly imperfect signal of species diversification we recover

here. Ultimately, we believe that it is better to present both sets of curves as they stand, given there is still no robust linkage in the literature between genus and species-level dynamics, and acknowledge their imperfections clearly in the discussion. To aid in this, we have made explicit reference to the issue in the caption for fig. 3 (L1056–1058) and direct the reader to the longer discussion of this issue in the text (L340–347).

Lines. 144-169: It's a bit hard to read here. Why don't you describe the diversity dynamics in the eight regions in one paragraph. For instance: "Declines in genus and species richness occurred during the interval containing OAE2 in all four," At first glance, I was confused, what are the four regions? I can only find the answer (East Pacific, Tethys, WIS and Atlantic and Gulf) when I read the whole paragraph.

We agree that our original phrasing did not aid with manuscript clarity here. We now describe the dynamics in one paragraph as the reviewer suggests (L144-169). Our original structure was designed to highlight how regions away from the Global North depart from the 'global' diversity signal, so we have tried to retain this in the revised structure as we view it as a key result in our study.

Line 353. "Middle" => "Early".

Change made (L401)

Line 577-578: Do you mean five superfamily and twelve suborders? Please check.

Quite right, these were misordered, change made (L625)

I tried several times and can't access the code via the link provided above.

We apologise for this issue. We have revised our ESM in light of this review and have made this new version available via the manuscript submission portal using Nature's FigShare service. We hope that this has addressed the problem, but if not then please get the editor to contact us and we will endeavour to make the full ESM available in another way

Reviewer #2:

Spatiotemporal variations of biodiversity impact our understanding of past evolution. This manuscript focuses on the debate of what drives diversification and whether we can faithfully identify these drivers from the fossil record. I do not have major concerns on the technical part of the paper. However, the author(s) soundly criticized past studies and, in some sentences, all palaeontologists. I am not sure if this is the best solution to promote this paper, but this is up to the authors.

We are pleased that the reviewer views our paper as technically sound. We agree that our original phrasing could be interpreted as a little damning of all palaeontologists, which is certainly not our intention. We have modified our phrasing to reduce this possibility. We instead view the need for spatially sensitive analysis as the 'promoting' point for the work.

From an absolutely objective perspective, most researchers need a simple solution for many applications of biodiversity changes e.g., in palaeoenvironment studies. The approach provided by the author(s), though impressive and better, is not simple, nor a solution to the general problem. Nevertheless, this is a well-written manuscript that will add new insights into our understanding of drivers of past extinction, recovery, and diversification.

We thank the reviewer for their positive perspective on the contributions of our manuscript to the field. We agree that our analytical strategy is not the most straightforward to implement, but this is largely because we aimed to analyse rates rather than standing diversity. Measuring the latter in a spatially sensitive framework is much easier to achieve using standard analytical tools and R packages and is a more achievable solution for future workers. We view our take home message as the need for spatially sensitive analysis to effectively tackle questions of extinction, recovery and diversification in the fossil record.

Line 69-71: True, but the words are rather harsh.

We have revised our phrasing to 'some palaeontologists' (L69). This is a more accurate reflection of the state of the field and implicitly acknowledges the excellent efforts of other palaeontologists to move away from global analyses towards spatially sensitive frameworks.

Line 137: WIS first occurred here. But the abbreviation was only explained in line 421.

We have ensured that the abbreviation is now provided at the first instance of the complete phase (L95).

Lines 144-155: When reading this paragraph, the question popped up: Would the bias of sampling efforts completely rule out?

Our analytical strategy is designed to minimise various sources of sampling bias as much as possible, but we agree that it is probably impossible to eliminate its effects entirely. We do not address this point within the lines referenced by the reviewer to keep the results free from any discussion but have referred to the issue of residual sampling bias as a new section later in the manuscript (L324–340). We believe that this inclusion greatly improves the quality of our work and thank the reviewer for suggesting that we consider the issue of sampling in more detail

Lines 242-243: the sentence "regression-driven loss of habitat space is more commonly associated with elevated extinction rates" needs references.

We have added references to support the impact of regression on extinction, but we have also caveated that this relationship is not necessarily consistent across space, which helps to clarify why rising sea levels might instead coincide with elevated extinction rates (L243–250).

Lines 248-250: difficult to follow, explain why prevalence in the smaller sampling regions and absence in the larger occur.

This was a remark on the prevalence of significant correlations between sp/ex and ammonoid diversity in the smaller sampling regions, so we have added a reference to the figure and reworked the phrasing for clarity. We have also added a comment with references on a possible explanation for this finding based on the literature – that RQ effects are overwhelmed by CJ effects at larger spatial scales (L260–264).

Lines 250-252: regional ammonoid origination and extinction rates showed marked heterogeneity. How much difference can be inferred as marked heterogeneity? Please explain.

We have removed 'marked' as this phrasing was vague with no clear basis for determining if difference is marked or otherwise (L265).

Lines 288-293: Not very clear here. Taxonomic assignments are not affected by measurement location

We were a little uncertain of whether the reviewer is giving their interpretation of what we said in the manuscript in relation to the next point raised, or simply stating the above point plainly. We aimed to make the same point in the manuscript (L304–305), so we hope that we have understood the reviewer here and that our response below addresses their intended concern.

Lines 288-293 (cont): This seems to contradict the idea of spatial heterogeneity mentioned earlier. And the following sentence “fossil occurrences associated with taxonomic assignments display geographically biased sampling” doesn’t seem to make sense. Explain in more detail and give some references.

We disagree that independence of taxonomic assignment from spatial location contradicts the idea of spatial heterogeneity. Our point is that the sampling of fossils as data points is spatially biased, but that this spatial bias will not have any impact on the taxonomic identities assigned to those data points (L304–309). *Baculites* sp. identified in one location will belong to the same taxonomic unit as *Baculites* sp. from another location, even if one of those locations is disproportionately better sampled than the other. We have revised our phrasing to help ensure that this point is clear.

Lines 309-312: What then was the reason (driver) for ammonoid extinction?

Sentence and references added for the current stance in the literature on ammonoid extinction at the K-Pg (L354–358). We did this at a different point in the manuscript to the lines where the reviewer raises their comment to keep our conclusions as free as possible from inferences made elsewhere in the literature.

Line 311-312: the sentence “the hypothesis of long-term global diversity decline prior to their final extinction at the K-Pg boundary” needs references.

References added (L351).

Reviewer #3:

I laude the authors for their focus on regional diversity trends as a way to understand “global” diversification patterns of ammonoids during the Late Cretaceous. As Geerat Vermeij stated in 2003, global diversity does not really mean what it implies given the well-established biases in the known fossil record at the global scale. This study is an interesting and important contribution to the understanding of a major group of animals as they approached their ultimate demise. The results indicate that despite parsing out global data into distinct regions, ammonoid diversity was not waning during the Late Cretaceous as many workers have suggested. Moreover, the results suggest that drivers of regional diversity patterns are more likely driven by regional drivers, such as temperature and sea-level change.

We are pleased by reviewer’s positive response on our work and thank them for drawing our attention to Vermeij (2003), whose point of whether global trends are actually meaningful is very pertinent to our work, so we have added this reference to the introduction (L65).

Overall, this study highlights an exciting approach for assessing the evolutionary history of Late Cretaceous ammonoids and potential application to other major invertebrate groups in the fossil record. The methods also seem sound to me, particularly in their approach for controlling sampling biases. I only have a few minor comments to make regarding the discussion and methodology, otherwise I believe this manuscript should be accepted once these issues have been addressed.

We are pleased that the reviewer considers our approach to be robust and thank them for their additional comments, which we found to be very thought provoking.

Minor comments:

Sea-level change has long been attributed to be a driver of both extinction and origination in dynamics in the fossil record. However, evidence suggests that sea-level change itself does not meaningfully effect macroevolutionary dynamics, regardless of spatial scope (Holland 2012, Paleobiology). Given that ammonoids are pelagic animals and even less likely to be affected by sea-level change, I fail to see the reason to include sea-level analysis in this study. Sea-level change will have a stronger relationship with the types of facies preserved, with relatively deeper-water facies being deposited during times of sea level rise, which are more likely to preserve ammonoid fossils than shallower water facies. Which leads into my next comment.

We included sea level due to its effect on the extent of shallow marine shelf area. While ammonites are pelagic, we nonetheless expect increased volume of pelagic habitat overlying productive shallow marine shelf area to promote their diversification. Additionally, many heteromorph ammonitez, which became increasingly diverse through the Late Cretaceous and are well represented in our dataset, were nekto-benthic rather than pelagic and so shallow marine shelf area will be even more relevant to these taxa. We chose to use sea level as a proxy rather than a direct measurement of shelf extent due to ongoing uncertainty between palaeogeographic reconstructions, which would only be compounded by the uncertainty in sea level change itself.

Holland (2012) did find that that the relationship between sea level and shallow marine area was complex and asserted that at local scales sea level will not be a simple predictor for diversity under the species area effect. Holland (2012) also showed, however, that at regional scales that sea level and shallow marine area remain positively correlated (Fig. 4 in paper). Additionally, Holland (2012) did not actually conduct any formal analyses of empirical diversity versus changing area under sea level and simply predicted diversity as a function of benthic area using the Arrhenius equation. More recent studies of the empirical fossil record instead provide support for long term sea level as having predictive power at larger spatial scales (e.g., Roberts and Mannion 2019, *loc cit*).

Ultimately, we assert that sea level remains a worthwhile predictor to test, but we accept that its relationship with diversity may be complex. We have added new material to the discussion to properly address this possibility (L241–250). At best, sea level had a genuine effect and is captured by our analyses. At worst, its effects could be ignored as excluding it from the PyRateMBD analyses would not impact the modelled effects of the other predictor variables.

Again, while I appreciate the effort of the authors to consider the regional heterogeneity of diversity dynamics, there is no mention regarding the nature of the stratigraphic record at the regional scale. While ammonoids are preserved across the majority of marine facies, they are most likely to be live and preserve in relatively deeper-water environments (80–400m; Ritterbush 2014, *J. of Zoology*; Lukeneder et al. 2010, *Earth Planet. Sci. Lett.*). As such, the structure of the stratigraphic record regionally would have a greater effect on diversity rather

than sampling biases. I wonder if stratigraphic packages regionally would potentially have a 1st order control on diversity. I do realize, however, the difficulty in actually testing this, given the poor state of facies/deposition environment data in the PBDB and lack of global stratigraphic packages in Macrostrat. Just a thought.

It has been previously determined that PyRate provides robust results even when confronted by strong preservation rate heterogeneity. As such, our regionalised results will account for preservation rate variation induced by regional stratigraphic architecture. We still agree, however, that there will be a trade off between biological signal versus stratigraphic noise in relation to region size. As the reviewer points out, this would be challenging without proper stratigraphic data for each region, but it is still worth pointing this out in the manuscript. No work has been done on how facies architecture might bias PyRate analyses and this will certainly be important to tackle in future work, particularly given the reviewer's pertinent point regarding sea level controlling facies types. We have added a paragraph to the discussion to outline these issues and believe that it significantly improves the presentation of our study limitations (L324–340).

Lastly, while sea-surface temperatures might have mixed effects on diversity in this study, it is possible that temperature in conjunction with oxygen might be a better environmental component to assess. While modern cephalopods are sensitive to temperature, they have behavioural adaptations to compensate for increased temperatures, but not for temperature-dependent hypoxia.

As with the previous issue of the lack of stratigraphic databases to test for facies-driven biases, our temperature and redox data are unfortunately insufficient to properly tackle this issue as they are one dimensional time series which summarise global environmental states, rather than providing the necessary information on how temperature and hypoxia spatially co-varied between regions. A couple of studies have convincingly demonstrated this effect in the fossil record, but all have spatially resolved estimates of oxygen availability and temperature from earth system models coupled with metabolic-limitation models of marine invertebrates. Such analyses fall beyond the scope of this study, but they also provide excellent examples of spatially explicit approaches to modelling diversification which can additionally solve the issues we raise regarding spatially biased proxy records. We have added reference to these model-based studies in our discussion to provide a clear pathway for future work to pursue (L390–392), as well as using the empirical point of temperature-dependent hypoxia to present the potential limitations of our study more thoroughly (L251–258), although recent meta-analysis of cephalopod temperature tolerances found consistently negative effects of warming across lineages and ontogenies (Borges et al., 2023, *loc cit*).

Otherwise, great and interesting study!

Thanks again!